# Comparison of the Disease Severity and Outcome of Vaccinated COVID-19 Patients with Unvaccinated Patients in a Specialized COVID-19 Facility: A Retrospective Cohort Study from Karachi, Pakistan

**DOI:** 10.3390/vaccines11071178

**Published:** 2023-06-30

**Authors:** Muneeba Ahsan Sayeed, Elisha Shalim, Fizza Farooqui, Shaiza Farman, Maheen Khan, Anika Iqbal, Ishfaque Ahmed, Abdul Wahid Rajput, Abdul Razzaque, Saeed Quraishy

**Affiliations:** 1Sindh Infectious Diseases Hospital & Research Center, Karachi 75300, Pakistan; elisha10.es@gmail.com (E.S.); fizza.jameel@gmail.com (F.F.); shaizafarman@gmail.com (S.F.); khan.maheen3011@gmail.com (M.K.); anikaiqbal.7432@duhs.edu.pk (A.I.); ishfaqueahmed.7254@duhs.edu.pk (I.A.); wahid.rajput@duhs.edu.pk (A.W.R.); abdul.razzaq@duhs.edu.pk (A.R.); vc@duhs.edu.pk (S.Q.); 2Infectious Diseases Department, Dow University of Health Sciences, Karachi 75300, Pakistan

**Keywords:** COVID-19, vaccinated, unvaccinated, comparison, outcome, hospitalized

## Abstract

We compared the clinical characteristics and outcome of vaccinated hospitalized COVID-19 patients with unvaccinated hospitalized COVID-19 patients. A retrospective cohort study was conducted at the Sindh Infectious Diseases Hospital and Research Center, Karachi, Pakistan. A total of 1407 hospitalized COVID-19 positive patients were included from April 2021 to March 2022, of which 812 (57.71%) were males. Of the 1407, 378 (26.87%) patients were vaccinated while 1029 (73.13%) were unvaccinated. Of the vaccinated patients, 160 (42.32%) were partially vaccinated while 218 (57.68%) were fully vaccinated (vaccine breakthrough infection). Fewer unvaccinated COVID-19 patients survived compared to vaccinated patients (62.5% vs. 70%, RR 0.89, 95% CI: 0.82–0.96, *p*-value = 0.004). Despite there being more vaccinated patients above 60 years of age (60.05% vs. 47.13%), their risk of mortality was lower by 43% (OR = 0.578; CI = 0.4201 to 0.7980, *p* = 0.0009). On survival analysis, vaccinated patients had better 30-day survival compared to unvaccinated patients (*p* = 0.028). Moreover, comparing waves 3–5, unvaccinated patients of wave 4, which was driven by the delta variant, had the worst survival (51.8%, *p* ≤ 0.001) while vaccinated patients of wave 3 (driven by the alpha variant) had the best survival (71.6%).

## 1. Introduction

The COVID-19 pandemic has ended but its outbreaks are still prevalent globally [1]. Pakistan is the world’s fifth most populous country and is one of those seven countries whose strategies for combating COVID-19 were praised by the World Health Organization. In February 2020, Pakistan witnessed its first case of COVID-19. In April 2020, the National Command and Operation Centre (NCOC) was established by the Government of Pakistan to govern and monitor the national COVID-19 efforts with a holistic approach. To date, six waves of COVID-19 have been reported from Pakistan, driven by different COVID-19 variants. By the end of April 2023, over 1580,760 confirmed cases and more than 30,000 deaths had been reported, which is the world’s 29th highest death toll [2]. COVID-19 vaccination played a crucial part in reducing the spread of COVID-19. Globally, 70% of the population has received a COVID-19 vaccine and the vaccine coverage in Pakistan is 59.5% [1,3]. Despite vaccination, breakthrough infections are prevalent, as evident by the recent surges in China and Europe [4].

The primary focus has been on the evaluation of SARS-CoV-2 vaccines in preventing hospitalization and symptomatic infections [5,6,7,8,9]. Hospitalized COVID-19 patients can advance to critical disease, including organ failure and death. However, vaccinated patients, on developing breakthrough COVID-19 infection, initiate memory antibody and cellular responses, which could alleviate disease progression, possibly averting life-threatening organ failure and death [10].

There are different types of COVID-19 vaccines and in Pakistan the available vaccines are viral vector vaccines (AstraZeneca, CanSinoBio, Pakvac and Sputnik V), mRNA vaccines (Pfizer and Moderna) and inactivated vaccines (Sinopharm and Sinovac). As COVID-19 vaccines primarily target the spike protein, the alterations in this region may reduce the capability of the immune system to recognize and neutralize the virus effectively [11]. Thus, breakthrough infections are associated not only with the efficacy of vaccines but also with the variant causing COVID-19 [12]. Each vaccine has a different efficacy against different COVID-19 variants. In Pakistan, the COVID-19 vaccination drive commenced in February 2021, starting with frontline health care workers, followed by the elderly (>65 years of age) in March 2021. A free of charge vaccination service through helpline number 1166 was initiated in which registration was carried out through National Identity Card numbers. After registration, a text message was sent on the cell phone with a code, date of appointment and name of the assigned nearby vaccination center from which the recipient could receive the vaccine. More than 2000 mass vaccination centers were established throughout the country that worked round the clock. Immunization certificates could then be downloaded from the National COVID-19 Immunization Portal (https://nims.nadra.gov.pk; accessed date 15 June 2023) or by visiting local centers of NADRA (National Database and Registration Authority) [13,14]. With public–private partnership, drive-through COVID-19 vaccinations points were also developed to assist the masses and ease the burden on vaccination centers established in hospitals and other public facilities. Vaccination drives were launched at a time when Pakistan had already witnessed two COVID-19 waves and wave 3 was right around the corner. Wave 3 in Pakistan started in March 2021 and was driven predominantly by the alpha variant, wave 4 started in July 2021 and was driven by the delta variant, while wave 5, which lasted from November 2021 to March 2022, was driven by the omicron variant [15,16]. In this study, we compared vaccinated hospitalized COVID-19 patients with unvaccinated hospitalized COVID-19 patients in terms of disease progression and outcome. We also assessed the response of vaccination against different variants specific to each COVID-19 wave.

## 2. Materials and Methods

A retrospective cohort study was conducted on 1407 hospitalized COVID-19 patients at the Sindh Infectious Diseases Hospital and Research Center (SIDH&RC)/Dow University of Health Sciences, a 175-bed tertiary care designated infectious diseases hospital located in Karachi, Pakistan. We sought ethical approval from the Institutional Review Board of Dow University of Health Sciences (IRB 2339/DUHS/Approval/2021/671). Since data were collected retrospectively, the requirement of informed consent was waived by the Institutional Review Board. All admitted patients over 16 years of age who tested positive for COVID-19 on reverse transcriptase polymerase chain reaction (RT-PCR) assay through nasopharyngeal swab, presenting to the hospital from April 2021 to March 2022, were included. Patients younger than 16 years, or those who were not managed primarily at SIDH&RC, were excluded. We collected data from the hospital medical records and HMIS system by chart review of each patient’s demographics, medical history, disease severity, vaccination status, name of received vaccine, disease progression, intensive care unit (ICU) stay, treatment and outcome. To preserve confidentiality, each patient was coded with a study-specific number and personal information including names and other identifiers was removed. Only the authors had access to the collected data.

The population involved in the study was divided into two groups: vaccinated and unvaccinated, based on the COVID-19 vaccination status. Patients were considered “fully vaccinated” if they had received a COVID-19 vaccine of the recommended dosage, which is two doses for Sinovac, Sinopharm, AstraZeneca, Pfizer, Moderna and Sputnik while a one dose regimen for CanSinoBio and Pakvac as part of primary immunization. “Partially vaccinated” were those who had received only one COVID-19 vaccine dose out of a series of two at the time of admission. Unvaccinated patients were those who did not receive any COVID-19 vaccine. Data for COVID-19 wave 3 was collected from April 2021 to June 2021, for wave 4 from July 2021 to October 2021, while, for wave 5, from November 2021 to March 2022.

Disease classification of COVID-19 was according to the then national COVID-19 treatment guidelines. Mild COVID-19 was defined as having symptoms of COVID-19 such as fever, cough, sore throat, fatigue, headache, myalgia, nausea, vomiting, diarrhea and loss of taste and smell with no shortness of breath and normal oxygen saturation (SpO2) in room air. Moderate COVID-19 was defined as SpO2 ≤ 94% but >90% with mild pneumonia. Severe COVID-19 was defined as SpO2 ≤ 90% in room air with >50% lung infiltrates on chest radiograph while critical COVID-19 was defined as having acute respiratory distress syndrome (ARDS), multi-organ dysfunction or septic shock with the patient requiring non-invasive or invasive ventilation.

The collected data were analyzed through IBM Statistical Package for the Social Sciences (SPSS), version 24.0 (https://www.ibm.com/support/pages/downloading-ibm-spss-statistics-24; accessed date 30 April 2022). For continuous data, we report the mean and standard deviation/median with interquartile range depending on the normality assumption. Frequency and percentages are reported for categorical data. To compare the demographics and outcome between vaccinated and unvaccinated COVID-19 patients, a chi-square test was applied and a *p*-value of <0.05 was considered significant. Computation was carried out by cross-tabulation to generate the odds ratio (OR) and relative risk (RR) with 95% confidence intervals (95%CI) for comparative and outcome data. We compared the 14-day and 30-day mortality between vaccinated and unvaccinated COVID-19 patients. Kaplan–Meier survival curves were then plotted for both 14-day and 30-day survival. We also compared the 30-day survival of vaccinated and unvaccinated COVID-19 inpatients of waves 3, 4 and 5 using log-rank survival analysis and plotted it using Kaplan–Meier survival curves.

## 3. Results

### 3.1. Baseline Characteristics

Records of 2163 patients were available from April 2021–March 2022 of which 756 were excluded due to missing data, thereby leaving 1407 records for analysis. Of the 1407 patients, 812 (57.71%) were males. Based on COVID-19 vaccination status, most of the patients [1029 (73.13%)] were unvaccinated while 378 (26.87%) were vaccinated against COVID-19. Among vaccinated patients, 160 (42.3%) were partially vaccinated while 218 (57.7%) were fully vaccinated as shown in Figure 1. None of them received a COVID-19 booster vaccine because the COVID-19 booster vaccination roll out started only in January 2022 in Pakistan.

Of the vaccinated patients, 303 (80.1%) received inactivated vaccines (Sinopharm 194, Sinovac 105, Pakvac 4), 45 (11.9%) received adenovirus vaccines (CanSino 35, AstraZeneca 10), while only 18 (4.76%) received mRNA vaccines (Pfizer 13, Moderna 5).

Of the 1407 patients, 950 (67.5%) had comorbidities of which hypertension was the predominant one, seen in 675 (48%) patients, followed by diabetes mellitus and ischemic heart disease. Based on disease severity, the majority [820 (58.3%)] of patients had severe COVID-19 on admission, of which 184 (22.4%) were vaccinated while 636 (77.6%) were unvaccinated. While 243 (17.3%) had moderate COVID-19, 232 (16.5%) had mild COVID-19 and only 112 (8%) patients had critical COVID-19. Of the total number of patients, nearly half of the patients received tocilizumab, with the majority being in the unvaccinated group, while almost all patients received steroids as shown in Table 1.

### 3.2. Comparison of Baseline Characteristics of Unvaccinated COVID-19 Patients with Vaccinated COVID-19 Patients

On comparison, unvaccinated COVID-19 patients, despite being younger (mean age 60 vs. 65 years, *p*-value < 0.0001) and having fewer comorbidities (63.9% vs. 77.2%, OR 0.52, 95% CI: 0.39–0.68, *p*-value < 0.0001), had increased risk of severe COVID-19 at the time of admission (61.8% vs. 48.7%, OR 1.7, 95% CI: 1.35–2.16, *p*-value < 0.0001) than vaccinated patients. Moreover, unvaccinated patients were more likely to receive tocilizumab (50% vs. 34%, OR 1.95, 95% CI: 1.5–2.5, *p*-value ≤ 0.0001) and steroids (94% vs. 83%, OR 3.22, 95% CI: 2.2–4.7, *p*-value < 0.0001) as shown in Table 2.

Of the 1407 patients, 619 (44%) were admitted in COVID-19 wave 3, 628 (44.6%) in wave 4, while 160 (11.4%) were hospitalized in wave 5 of COVID-19. Among 619 wave 3 patients, 67 (10.8%) were vaccinated while 552 (89.2%) were unvaccinated. In wave 4, 151 (24%) were vaccinated while 477 (76%) were unvaccinated but in wave 5 all 160 (100%) admitted patients were vaccinated against COVID-19.

### 3.3. Comparison of Outcome of Unvaccinated COVID-19 Patients with Vaccinated Patients

Of the 1407 patients, 84 patients were excluded at the time of assessing outcome as they were either referred to some other hospital or had left the hospital against medical advice. To compare outcomes, 1323 patients were considered. Table 3 is comparing the outcome of unvaccinated COVID-19 patients with vaccinated patients. On comparison, CRS was more often seen in unvaccinated COVID-19 patients compared to vaccinated COVID-19 patients (34.1% vs. 25.1%, RR 1.36, 95% CI: 1.12–1.66, *p*-value = 0.002). Meanwhile, mechanical ventilation, ICU stay, complications and disease progression were the same in both groups. However, disease progression to death was more common in unvaccinated patients compared to vaccinated patients (69.8% vs. 57.9%, RR 1.21, 95% CI: 1.06–1.39, *p*-value = 0.004).

Overall in-hospital mortality was 35.4%. Fewer unvaccinated COVID-19 patients survived compared to vaccinated patients (62.5% vs. 70%, RR 0.89, 95% CI: 0.82–0.96, *p*-value = 0.004). Despite there being more vaccinated patients above 60 years of age (60.05% vs. 47.13%), their risk of mortality was lower by 43% (OR= 0.578; CI= 0.4201 to 0.7980) compared to unvaccinated COVID-19 patients, which was statistically significant (*p*-value = 0.0009). Moreover, among vaccinated patients, mortality was lower in completely vaccinated patients compared to partially vaccinated patients (40.7% vs. 45%, OR 0.84; CI: 0.55–1.28).

The median (IQR) length of hospital stay was 7 (4–12) days in unvaccinated COVID-19 patients compared to 6 (4–11) days in the vaccinated group. In severe COVID-19 patients, the risk of prolonged hospitalization was 35% less in vaccinated patients as compared to unvaccinated patients (OR 0.65; CI: 0.52–0.83, *p*-value = 0.0004). Moreover, in moderate COVID-19 patients, the unvaccinated patients had a 1.5 times higher risk of disease progression to death compared to vaccinated patients (20% vs. 9.72%, OR 1.58, CI: 0.93–5.94). In patients with comorbidities, COVID-19 vaccination significantly lessened the risk of mortality compared to unvaccinated patients (OR 0.71; CI: 0.53–0.96, *p*-value = 0.03).

### 3.4. Survival Analysis

Figure 2 illustrates the Kaplan–Meier 14-day and 30-day survival curves for hospitalized patients with COVID-19, stratified by unvaccinated versus vaccinated. In the vaccinated group, 253 (66.9%) survived for 30 days, while 621 (60.3%) in the unvaccinated group had 30-day survival. Censored patients were those who were lost to follow up and there were 22 (3.78%) in the vaccinated group and 61 (5.9%) in the unvaccinated group. It is found that 14-day survival was slightly better in the vaccinated group but it was not statistically significant (log-rank test value, *p* = 0.057), while 30-day survival was significantly better in the vaccinated patients (log-rank test value, *p*-value = 0.028) compared to unvaccinated patients (Figure 2a,b).

Figure 3 shows the Kaplan–Meier 30-day survival curve for hospitalized COVID-19 patients, stratified by vaccination status and COVID-19 waves (waves 3–5). It is found that the vaccinated patients of COVID-19 wave 3 had the best (71.6%) while unvaccinated patients of COVID-19 wave 4 had the worst (51.8%) survival rate. Moreover, vaccinated patients of COVID-19 wave 4 had a lower survival rate (65.6%) compared to vaccinated patients of wave 3 (71.65%) and wave 5 (66.3%).

## 4. Discussion

In this study, we compared the clinical characteristics and outcome of unvaccinated COVID-19 patients with vaccinated hospitalized COVID-19 patients. We also compared the survival of patients in COVID-19 waves 3–5 stratified by vaccination status. We found that the majority of the patients who were admitted with COVID-19 were elderly, had comorbidities, were unvaccinated and presented with severe COVID-19 on admission.

Most of the admitted patients had one or more comorbidities. Several studies have shown that, along with the COVID-19 severity, mortality is also associated with underlying medical conditions [17]. In one meta-analysis, hypertension was the most predominant comorbidity associated with COVID-19 [18]. Meanwhile, in other studies, cardiovascular diseases and diabetes mellitus were the most common comorbidities [19,20]. The finding of our study correlates with other studies from Pakistan in which hypertension and diabetes mellitus are the predominant comorbidities [21].

Most of the patients in this study were unvaccinated despite the availability of free COVID-19 vaccines. Statistical data revealed that a significant proportion, two out of five individuals, exhibit hesitancy regarding receiving COVID-19 vaccines [22]. Based on a Gallup COVID-19 tracking survey, a substantial percentage (49%) of individuals in Pakistan expressed hesitancy towards COVID-19 vaccines [23]. Moreover, another survey reported that 50.6% of the participants exhibited reluctance towards the vaccine. The study attributed this hesitancy to concerns regarding potential side effects such as autism, infertility and autoimmune disorder, false perception of natural immunity against COVID-19 and even mortality [24]. To combat this, the Government of Pakistan started an awareness campaign for citizens by advertisements, leaflets, online flyers, calling ringtones and awareness videos in six different spoken languages in the country. Moreover, celebrities, religious scholars, leaders and media personnel were engaged to spread awareness of the importance of vaccination in order to increase vaccine uptake and adherence to COVID-19 preventive measures. In addition, COVID-19 vaccination was made mandatory for local travel, workplaces and public and private gatherings. The conversion of the unvaccinated population to vaccinated thus took place at a slow but steady pace and, by the end of wave 5, most of Pakistan’s population was vaccinated, as evident in this study in which all wave 5 COVID-19 patients were vaccinated. As of June 10, 2023, 339,286,324 doses of COVID-19 vaccines have been administered in Pakistan [25].

In this study, the majority of the vaccinated patients received inactivated vaccines such as Sinopharm followed by Sinovac because these were the readily available vaccines then. When the world was facing constraints in getting COVID-19 vaccines, Pakistan initiated its vaccination drive after getting 500,000 doses of Sinopharm from China [26]. Later, other vaccines were procured, including AstraZeneca, CanSinoBio, Sputnik V, Pfizer and Moderna [27].

This study reiterates that COVID-19 vaccines help in preventing disease progression and death. It was interesting to see that vaccinated COVID-19 patients, despite being older and having more comorbidities, had fewer severe COVID-19 cases on admission with a lower mortality rate compared to unvaccinated COVID-19 patients. Papaioannou et al. have reported similar results in their study. They found that vaccinated patients were older than unvaccinated COVID-19 patients, yet had a better outcome and a lower mortality rate (HR: 2.59, 95% CI: 1.69–3.98, *p* < 0.0001) [28].

In our study, the unvaccinated patients were relatively younger than the vaccinated patients were. This is likely because the vaccination program in Pakistan started with the vaccination of the elderly (>65 years) population in early 2021. This could explain why the majority of the COVID-19 inpatients consisted of unvaccinated younger people (median age 60); the few vaccinated patients who were hospitalized were either older (median age 65) or had several comorbidities. Ruiz et al. and Olivier et al. showed similar trends in their studies in Spain and France, respectively [29,30].

Mortality was higher in the unvaccinated group compared to the vaccinated group as reported in other studies. Most of the patients in the vaccinated group who died either were more than 60 years of age or had one or more comorbidities. We found that mortality and disease severity were the two major outcomes that were affected by vaccination status. However, unlike other studies from India and the US, factors such as ICU stay, mechanical ventilation and disease complications were the same in both groups. Data from India showed that disease severity, hospital stay, requirement of invasive and non-invasive ventilation and mortality were considerably less in vaccinated COVID-19 inpatients compared to unvaccinated patients [31]. Similarly, in the US a case–control study conducted at 21 hospitals in 18 states showed that vaccination with an mRNA COVID-19 vaccine reduced the risk of COVID-19-related hospitalization, invasive mechanical ventilation and mortality. These findings favor the fact that vaccination lessened the risk of progression to severe COVID-19 in vaccine breakthrough infections when compared to no vaccination [23]. Tenforde et al. also found that vaccinated patients compared to unvaccinated patients had a lower risk of requiring not only invasive mechanical ventilation (7.7 % vs. 23%), but also are less likely to require intensive care (25% vs. 40%) and had a lower mortality rate (6.3% vs. 8.6%) [32].

Among vaccinated patients, the 30-day mortality rate was 16.4% in wave 3, 29.8% in wave 4 and 29.4% in wave 5. Despite vaccination, in Pakistan mortality was high in waves 4 and 5, which were driven by delta and omicron variants, respectively. Studies have shown that vaccine effectiveness was 77.8% against delta and 55.9% against omicron variants compared to 88% against the alpha variant. Therefore, booster vaccinations are required which are considered to be more effective against delta (VE 95.5%) and omicron variants (VE 80.8%) [33]. This signifies that vaccination status is not the only factor determining COVID-19 outcome but factors such as comorbidities and variant type also play an important role.

Our study has some limitations; first, data were collected retrospectively, leading to potential information bias. Second, we did not perform gene sequencing for COVID-19 variants in all PCR samples; however, the specimens for which it was performed showed variants that were consistent with the country’s wave-specific COVID-19 variants. Third, data were non-generalizable as this was a single-center cohort study conducted at a specialized center for COVID-19 in Karachi. Fourth, the outcome was limited to discharge of patients and long-term effects of COVID-19 were not assessed. Fifth, we did not have data of prior COVID-19 infection status of these admitted patients and so cannot comment on the effects of COVID-19 reinfection on vaccine efficacy.

Since our study demonstrates how vaccination reduces the severity and improves overall survival through retrospective analysis alone, future research on antibody levels post-vaccination, metabolomics analysis and detailed information including past exposure to the virus is needed. This may help in obtaining more focused results and thus even pave the way for better development of vaccines and other treatment modalities.

## 5. Conclusions

In summary, this study compared the clinical characteristics and outcome of unvaccinated COVID-19 patients with vaccinated patients. Despite being elderly and having comorbidities, vaccinated patients had less severe COVID-19 infection on admission with a better 30-day survival compared to unvaccinated patients, while factors such as ICU stay, mechanical ventilation and complications were the same in both groups. Many factors along with vaccination status determine COVID-19 outcome.

## Figures and Tables

**Figure 1 vaccines-11-01178-f001:**
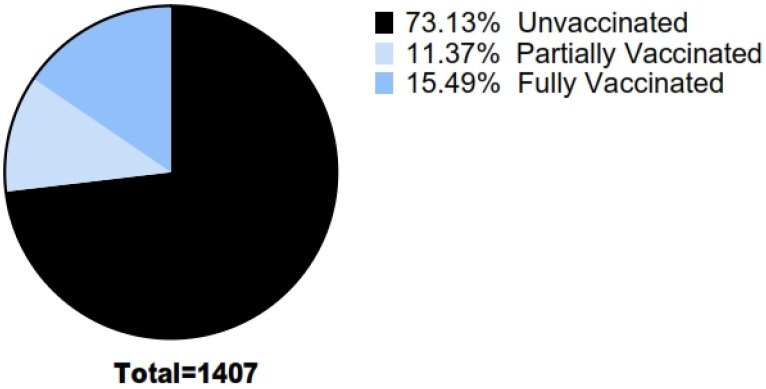
Proportion of Unvaccinated and Vaccinated COVID-19 Inpatients.

**Figure 2 vaccines-11-01178-f002:**
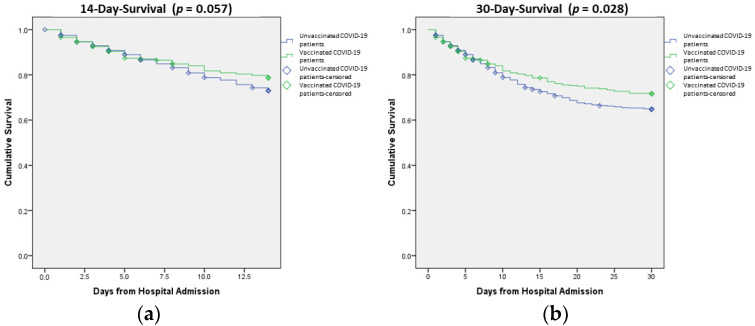
(**a**) Cumulative 14-day and (**b**) 30-day survival among vaccinated and unvaccinated hospitalized COVID-19 patients. The curve is illustrated with Kaplan–Meier method (log-rank test, 14-day (*p*-value = 0.057), 30-day (*p*-value = 0.028).

**Figure 3 vaccines-11-01178-f003:**
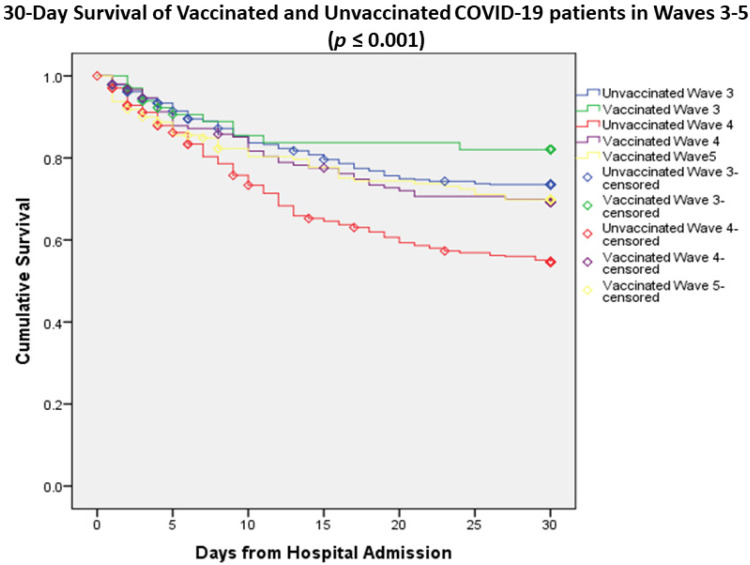
Survival analysis of COVID-19 waves 3–5 stratified by vaccination status.

**Table 1 vaccines-11-01178-t001:** Demographics and Baseline Characteristics of COVID-19 Inpatients.

Parameters	All *n* = 1407 (%)
Age, median (IQR)	60 (50–70)
Male	812 (57.7)
COVID-19 vaccination status	
Vaccinated Unvaccinated	378 (26.86) 1029 (73.13)
Comorbidities	950 (67.5)
Hypertension	675 (48)
Diabetes mellitus	548 (38.9)
Ischemic heart disease	148 (10.5)
Chronic kidney disease	35 (2.5)
Disease severity	
Mild COVID-19	232 (16.5)
Moderate COVID-19	243 (17.3)
Severe COVID-19	820 (58.3)
Critical COVID-19	112 (8)
Tocilizumab	646 (45.9)
Steroids	1286 (91.4)

**Table 2 vaccines-11-01178-t002:** Comparison of Demographics of Unvaccinated and Vaccinated COVID-19 patients.

Parameters	Unvaccinated COVID-19 Patients *n* = 1029 (%)	Vaccinated COVID-19 Patients *n* = 378 (%)	Odds Ratio [95% CI]	*p*-Value
Age, median (IQR)	60 (50–68)	65 (54–72)	0.57 [0.42 to 0.77]	<0.0001
Male	598 (58.1)	214 (56.6)	1.06 [0.84 to 1.35]	0.627
Comorbidities	658 (63.9)	292 (77.2)	0.52 [0.39 to 0.68]	<0.0001
Hypertension	468 (45.5)	207 (54.8)	0.69 [0.54 to 0.87]	0.002
Diabetes mellitus	382 (37.1)	166 (43.9)	0.72 [0.57 to 0.91]	0.022
Ischemic heart disease	56 (14.8)	92 (8.9)	0.56 [0.39 to 0.81]	0.002
Chronic kidney disease	24 (2.3)	11 (2.9)	0.79 [0.40 to 1.66]	0.328
Mild COVID-19	147 (14.3)	85 (22.5)	0.57 [0.42 to 0.77]	0.0003
Moderate COVID-19	166 (16.1)	77 (20.4)	0.75 [0.56 to 1.02]	0.0673
Severe COVID-19	636 (61.8)	184 (48.7)	1.70 [1.35 to 2.16]	<0.0001
Critical COVID-19	80 (7.8)	32 (8.5)	0.91 [0.59 to 1.38]	0.658
Tocilizumab	517 (50.2)	129 (34.1)	1.95 [1.53 to 2.5]	<0.0001
Steroids	970 (94.3)	316 (83.6)	3.22 [2.2 to 4.7]	<0.0001

**Table 3 vaccines-11-01178-t003:** Comparison of Outcome of Unvaccinated and Vaccinated COVID-19 patients.

Parameters	All *n* = 1323 (%)	Unvaccinated COVID-19 Patients *n* = 968 (%)	Vaccinated COVID-19 Patients *n* = 355 (%)	Relative Risk [95% CI]	*p*-Value
Cytokine release sydrome	419 (31.7)	330 (34.1)	89 (25.1)	1.36 [1.12 to 1.66]	0.002
Mechanical ventilation	310 (23.4)	226 (23.3)	84 (23.7)	0.99 [0.79 to 1.23]	0.942
ICU stay	538 (40.7)	398 (41.1)	140 (39.4)	1.04 [0.90 to 1.21]	0.613
Complications	765 (57.8)	562 (58.1)	203 (57.2)	1.02 [0.92 to 1.13]	0.802
Disease progression	685 (51.8)	507 (52.4)	178 (50.1)	1.05 [0.93 to 1.18]	0.495
Disease progression (*n* = 685) to death	457 (66.7)	354 (69.8)	103 (57.9)	1.21 [1.06 to 1.39]	0.004
Survivors	855 (64.6)	605 (62.5)	250 (70.4)	0.89 [0.82 to 0.96]	0.004
Non-survivors	468 (35.4)	363 (37.5)	105 (29.6)		

## Data Availability

Research data can be available on request to the corresponding author.

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
