# Peer review of "Comparison of the Disease Severity and Outcome of Vaccinated COVID-19 Patients with Unvaccinated Patients in a Specialized COVID-19 Facility: A Retrospective Cohort Study from Karachi, Pakistan"

_vaccines, 2023, doi:10.3390/vaccines11071178_

Round 1

Reviewer 1 Report

The authors compared the clinical characteristics and outcome of 1407 COVID-19 hospitalized patients in Pakistan between April 2021 and March 2022. The authors found vaccination value on severity of the disease and 30-day survival but did not see the merit of vaccination on ICU stay, mechanical ventilation and complications. Comments for the authors:

Major points:

1.      Figure 1: Please confirm the calculation. 160 should be 11.37% and 218 should be 15.49%. Also “Complete Vaccinated” should be “fully vaccinated”.

2.      Do the authors have data concerning COVID-19 infection status before or during vaccinations? I think this will affect the vaccine efficacy.

Minor points:

1.      The authors should use “COVID-19” throughout the manuscript for consistency or explain what doers “COVID” mean.

2.      Line 108: 378 should be 26.87%.

3.      Line 115: “Aztrazeneca” should be “Astrazeneca”.

4.      Line 116: “mRNa” should be ‘mRNA”.

Author Response

To,                                                                                       

The Editor-in-Chief & Reviewers,

Vaccines

Subject: Response to reviewer’s comments on revision of manuscript ID: 2452505

Thank you for considering the manuscript titled: “Comparison of the Disease Severity and Outcome of Vaccinated COVID-19-19 Patients with Unvaccinated Patients in a Specialized COVID-19 Facility. A Retrospective Cohort Study from Karachi, Pakistan” for peer review.

We have made the suggested revisions in the manuscript and point by point response is as follows:

  1. We noticed that your manuscript has a very high similarity rate with previously published material. During your revisions, please significantly reduce this similarity rate. This is extremely important.

We have revised the wordings of the manuscript to reduce the similarity index; however, the standard terms used in methodology could not be changed.

  1. We noticed that the main text of your manuscript is quite brief which may mean that the materials and methods, research background, future research directions, or possible applications of the research are not described in enough detail. Please consider the following points in your revisions: adding full experimental details, presenting completely all the results, and describing a comprehensive background to the research in the introduction section.

Added the requirements details.

Reviewer 1

The authors compared the clinical characteristics and outcome of 1407 COVID-19 hospitalized patients in Pakistan between April 2021 and March 2022. The authors found vaccination value on severity of the disease and 30-day survival but did not see the merit of vaccination on ICU stay, mechanical ventilation and complications. Comments for the authors: 

Major points:

  1. Figure 1: Please confirm the calculation. 160 should be 11.37% and 218 should be 15.49%. Also “Complete Vaccinated” should be “fully vaccinated”.

Thank you for pointing this out. Figure 1 updated

  1. Do the authors have data concerning COVID-19 infection status before or during vaccinations? I think this will affect the vaccine efficacy.

     No, we don’t have that data. We have mentioned this in study limitation. Line 335-337

Minor points:

  1. The authors should use “COVID-19” throughout the manuscript for consistency or explain what doers “COVID” mean.

Done. COVID replaced by COVID-19

  1. Line 108: 378 should be 26.87%.

     Thank you. Corrected. Line 154

  1. Line 115: “Aztrazeneca” should be “Astrazeneca”.

     Corrected line 164

  1. Line 116: “mRNa” should be ‘mRNA”

Thank you. Corrected line 165

Dr. Muneeba Ahsan Sayeed (Corresponding Author)

Reviewer 2 Report

This is an interesting article regarding the vaccination effect against COVID-19 disease in Pakistan. It is a retrospective Cohort study and this should be added in all relevant sections, including the title. An interesting point is that Sinovac and Sinopharm vaccines have been primarily used while mRNa Vaccines have been used relatively infrequently. Is should be mentioned what a complete vaccination is meant to be i.e two doses of the Sinovac or Sinopharm vaccines (primary immuniztion regimen). Details of whetherbany booster was administerd 4-6 months afterthe primary immunization regimen would be helpful. Indeed it would be helpful to cite that one dose regimens are the PakVac and CanSino 35 while the other vaccine regimens consist of two doses (Sinopharm, Sinovac, Astra zeneca, Pfizer and Modera mRNA vaccines) .  The reason for not vaccinated patients should be addressed i.e. is vaccination free in Pakistan? Was it relatively simple to get vaccinated? The age difference of 5 years in median terms cannot fully expalin why patients were not vaccinated as the non-vaccinated had significant comorbidities. Was vaccination voluntary, strongly advisable in Pakistan? 

Please minor editing as for example:

"Despite vaccination, break-through infection are prevalent, as evident by the recent surges in China and Europe [4]."

Please modify to infections

Author Response

To,                                                                                       

The Editor-in-Chief & Reviewers,

Vaccines

Subject: Response to reviewer’s comments on revision of manuscript ID: 2452505

Thank you for considering the manuscript titled: “Comparison of the Disease Severity and Outcome of Vaccinated COVID-19-19 Patients with Unvaccinated Patients in a Specialized COVID-19 Facility. A Retrospective Cohort Study from Karachi, Pakistan” for peer review.

We have made the suggested revision in the manuscript and point by point response is as follows:

Reviewer 2

This is an interesting article regarding the vaccination effect against COVID-19 disease in Pakistan. It is a retrospective Cohort study and this should be added in all relevant sections, including the title.

Added in title and methodology section

An interesting point is that Sinovac and Sinopharm vaccines have been primarily used while mRNa Vaccines have been used relatively infrequently. Is should be mentioned what a complete vaccination is meant to be i.e two doses of the Sinovac or Sinopharm vaccines (primary immuniztion regimen).

Mentioned in line 113-116

Details of whether any booster was administerd 4-6 months after the primary immunization regimen would be helpful.

None of the patients received booster vaccination as booster vaccination drive started in late January 2022. Mentioned in results section line 157

Indeed it would be helpful to cite that one dose regimens are the PakVac and CanSino 35 while the other vaccine regimens consist of two doses (Sinopharm, Sinovac, Astra zeneca, Pfizer and Modera mRNA vaccines) .

Added. Line 113-116

  The reason for not vaccinated patients should be addressed i.e. is vaccination free in Pakistan? Was it relatively simple to get vaccinated? The age difference of 5 years in median terms cannot fully expalin why patients were not vaccinated as the non-vaccinated had significant comorbidities. Was vaccination voluntary, strongly advisable in Pakistan? 

Discussed in details in Introduction (line 73-83   ) and discussion section (line 260-278)

Please minor editing as for example:

"Despite vaccination, break-through infection are prevalent, as evident by the recent surges in China and Europe [4]."Please modify to infections

Corrected (line 56)

Regards

Dr Muneeba Ahsan Sayeed (Corresponding author)